# Cooperative Communication Based Protocols for Underwater Wireless Sensors Networks: A Review

**DOI:** 10.3390/s24134248

**Published:** 2024-06-29

**Authors:** Muhammad Shoaib Khan, Andrea Petroni, Mauro Biagi

**Affiliations:** 1Department of Information, Electronics and Telecommunications (DIET) Engineering, Sapienza University of Rome, Via Eudossiana 18, 00184 Rome, Italy; muhammadshoaib.khan@uniroma1.it; 2Fondazione Ugo Bordoni (FUB), Viale America 201, 00144 Rome, Italy; apetroni@fub.it

**Keywords:** underwater wireless sensor networks, cooperative communication, cooperative error control, cooperative routing

## Abstract

Underwater wireless sensor networks are gaining popularity since supporting a broad range of applications, both military and civilian. Wireless acoustics is the most widespread technology adopted in underwater networks, the realization of which must face several challenges induced by channel propagation like signal attenuation, multipath and latency. In order to address such issues, the attention of researchers has recently focused on the concept of cooperative communication and networking, borrowed from terrestrial systems and to be conveniently recast in the underwater scenario. In this paper, we present a comprehensive literature review about cooperative underwater wireless sensor networks, investigating how nodes cooperation can be exploited at the different levels of the network protocol stack. Specifically, we review the diversity techniques employable at the physical layer, error and medium access control link layer protocols, and routing strategies defined at the network layer. We also provide numerical results and performance comparisons among the most widespread approaches. Finally, we present the current and future trends in cooperative underwater networks, considering the use of machine learning algorithms to efficiently manage the different aspects of nodes cooperation.

## 1. Introduction

Underwater wireless sensor networks (UWSNs) are receiving more and more attention due to their potential use in many applications, e.g., environmental monitoring, disaster prevention, resources investigation, scientific data collection and transmission [1]. Implementing UWSNs through wireless links represents the most convenient approach to achieve an efficient and cost effective communication framework. To this aim, several technologies have been considered, including acoustics [2], magnetic induction [3], radio-frequency (RF) and optics [4]. Among these, underwater acoustic communication (UWAC) has become a well-established solution since providing a balanced trade-off in terms of transmission distance, reliability and data rate. However, despite the link coverage provided by UWAC is in the order of hundreds of meters (up to kilometers), there are still many challenges related to signal propagation to be addressed. In fact, the underwater acoustic channel is time-varying in nature and characterized by multipath, rising from signal reflection at the water bottom and surface, and frequency selectivity [5] that impact on the received signal quality. Furthermore, the low speed of sound through the water medium causes long propagation delay that, together with the limited bandwidth available for acoustic systems, makes the achievable rate performance not competitive with that provided with other technologies.

Such impairments are typically investigated in the context of point-to-point UWAC, however they become relevant also in the more complex scenario of UWSNs. Figure 1 shows the typical architecture of a UWSN, which includes sensor nodes, sink nodes, and mobile nodes. Underwater nodes are the fundamental components of this network, and perform the collection of various oceanic data such as temperature, salinity, pressure, and so on. Such nodes are often battery-supplied and transmit information to the reference acoustic base station at the water surface via either direct link or an intermediate sink node that, based on capabilities, may gather data coming from multiple other nodes. Furthermore, mobile nodes like autonomous underwater vehicles (AUVs) and submarines may act as additional relay points between source and destination. Finally, the base station equipped with multiple interfaces performs the entire collection of underwater data and share them via RF wireless link to a land and/or satellite station, responsible for the required processing.

So, this architecture enables the effective collection, transfer, and application of ocean monitoring data through a complex interplay of numerous nodes and vehicles, both below and above the ocean.

As previously outlined, the effectiveness of a UWSN strictly depends on signal propagation issues that should be necessary tackled. Learning and borrowing solutions already employed in terrestrial networks [6], researchers have begun to explore the implementation of cooperative communications in order to improve the network performance. In the underwater context, nodes cooperation has been recognized as a promising strategy to mitigate the challenges rising from signal propagation, achieve energy efficiency and overcome the limits due to the bandwidth scarcity of acoustic systems and long propagation delay [7]. Few works in the literature investigate the theoretical modeling and performance of cooperative UWSNs [8,9], while many others concern feasibility studies related to specific scenarios and applications. For instance, the authors in [10,11] consider the implementation of cooperative UWANs to perform environmental monitoring and data collection, by also including AUVs as supporting mobile nodes. Nodes cooperation can be also fruitfully exploited for nodes positioning [12] and localization [13]. Finally, the use of cooperative approaches may led to significant improvements in terms of communication security [14,15], representing a fundamental challenge to be addressed also in underwater scenarios.

### 1.1. Motivation and Contribution

The paradigm of nodes cooperation was firstly introduced in terrestrial RF systems, defining that communication scenario where multiple nodes share their own transmit antenna to realize a virtual distributed multiple-antenna transmitter and benefit from spatial diversity to achieve improved performance in terms of data rate and reliability. Hence, nodes cooperation was only considered to improve the received signal quality. In RF communications the challenges introduced by channel propagation represent a well known and largely investigated subject. So, the procedures ongoing at the link and network layers are handled quite independently of what happens at the physical layer. Differently, the UWSNs work in a time-varying and hardly predictable scenario, where the channel impairments have a straight impact not only on pure signal propagation, but also on other aspects of network management. Therefore, researches have explored the potential of a cooperative approach to address many different issues. In fact, at the physical layer, the presence of cooperative nodes may help to improve the link reliability, mitigating multipath, attenuation and Doppler effects through adaptive schemes and signal processing. Another crucial challenge in UWAC is the long propagation delay, which has a significant impact when dealing bi-directional signaling between nodes. This is typically the case of error control, handled at the link layer by means of Automatic Repeat reQuest (ARQ) protocols. In this regard, involving cooperative nodes in the mechanism for ensuring a reliable data transmission seems to be promising for the achievement of a more efficient channel occupation as well. Lastly, nodes cooperation is also explored in the context of data routing at the network layer, aimed to not only optimize the data traffic, but also to minimize the nodes energy consumption and increase the network lifetime.

Based on these considerations, in this work we discuss how the paradigm of cooperative communications can be fruitfully exploited in the context of UWSNs, reviewing the state of the art of related strategies and protocols.

By investigating the most relevant issues about the physical, link and network layers of the protocol stack, herein we review:The signal combining techniques based on network nodes cooperation to mitigate the channel impairments;The ARQ protocols and medium access control (MAC) strategies recast in the context of cooperative UWSNs and employed for error control and channel resources management;The clustering and routing protocols tailored to cooperative UWSNs and aimed to traffic optimization and energy saving.

As a graphical support to the reading of the paper, Figure 2 reports at glance the classification of the cooperative techniques investigated in this work. A detailed analysis of the mentioned techniques is provided in the next sections. Some numerical results about the performance of the considered mechanisms and strategies are also provided. Finally, the recent trends in cooperative UWSNs management supported by machine learning are presented.

### 1.2. Paper Organization and Terminology

The paper is organized as follows. In Section 2 the physical layer mechanisms for diversity combining in cooperative communication are presented. Error and medium access control strategies for underwater cooperative scenarios are reported in Section 3. Network performance optimization through clustering and routing is discussed in Section 4. Performance discussion and some numerical results are provided in Section 5. Section 6 highlights the open issues and new research directions related to cooperative UWSNs supported by machine learning. Finally, conclusion is drawn in Section 7.

When dealing with physical layer issues, the amount of transmitted information is considered as a continuous flow of bits encoded in acoustic signals based on a certain modulation format. Passing to the link layer, information is formally organized in frames, composed of overhead and payload parts. At the network layer, each data unit is instead referred as packet and, following well known rules, is encapsulated in the payload of the link layer frame. Although the terms “frame” and “packet” related to different layers of the protocol stack have clear and different meanings, very often the term packet is indifferently used in both cases. Hence, in order to match with the most widespread terminology, in the rest of the paper the term packet will be used to identify the transmission data unit, but specifying whether referring to the link or network layer.

## 2. Physical Layer Cooperative Mechanisms

Providing a reliable underwater acoustic link is crucial to realize efficient UWSNs and, in general, an effective UWAC. In this regard, several studies in the literature have considered the paradigm of relay nodes cooperation in order to mitigate the channel impairments such as multipath and attenuation. The reference scenario is that one depicted in Figure 3, considering a first direct path between source and destination and a second one passing through a relay node.

Since the signal emitted by the source can be received by the relay as well, this latter can act as support to the destination by forwarding its own copy of the transmitted signal, exploitable for detection especially when the direct link has poor quality. The simplest mechanism adoptable by the relay is amplify and forward (AF), so it only amplifies the physical signal received from the source and forwards it to the destination. The authors in [16] present a cooperative AF scheme to mitigate the channel impairments. Specifically, the cooperation of the relay is exploited to turn the multipath into favorable conditions allowing a more reliable signal detection by the destination node. In [17], a more complex scenario with multiple relay nodes is considered, with AF combined with distributed space–time cooperative block coding (STCBC) to mitigate intersymbol interference (ISI) rising on the direct link and improve the communication reliability. Still related to a multi-relay configuration, in [18] relay selection and power loading for an orthogonal frequency division multiplexing (OFDM) based communication are investigated, with the goal to maximize the system capacity.

Overall, AF is simple and computationally efficient since concerning only the received signal amplification.

However, by doing so, together with the useful signal component, noise is amplified too. So, in the case of poor source-relay link quality, noise amplification may lead to the forwarding of a poor quality signal and a bad detection at the destination. As alternative, decode and forward (DF) overcomes the problem of noise amplification since the relay nodes performs detection and re-encoding before forwarding. Of course this mechanism improves the reliability of the forwarded signal, but at the expense of a higher computational and energy cost with respect to AF. Cooperative DF is considered in [19], where the authors investigate the theoretical performance and constraints related to a single-carrier communication suffering from ISI. In [20], a cooperative DF combined with OFDM is discussed, proposing a capacity criterion-based power allocation mechanism for performance improvement. A particular implementation referred as decode, interleave and forward (DIF) is proposed in [21], where turbo equalization, multiuser detection, and combining techniques are also realized to improve the link reliability while reducing the end-to-end delay. A deep performance analysis related to cooperative AF and DF OFDM based communication is given in [22,23], with authors demonstrating how relay node cooperation can be fruitfully exploited to improve the link performance from different points of view, including reliability, achievable data rate and outage probability. Finally, the authors in [24] propose a cooperative hybrid mechanism where, based on the channel quality, the relay node decides to perform either AF or DF.

Table 1 summarizes the state of the art about underwater cooperative communication, providing a general overview of the mechanisms employable at the physical layer. The nature of the underwater acoustic channel poses several challenges affecting the link performance in terms of reliability. The presence of a cooperative relay node allows the destination to benefit from the presence of an additional communication path where to receive a reliable signal when the direct link has poor quality. Despite a two-hop communication makes the overall channel effect partially attenuated with respect to a direct link case, it is still important to perform the most convenient processing at the relay node, that is AF or DF, based on the quality of the received signal. Moreover, due to the typical broadcast signal emission of underwater nodes, the destination may receive a copy of the same message from both the source and the relay. So, another issue to be deepened when dealing with cooperative communication is the synchronization of nodes transmission, in order to let the destination receive interference-free signals. The potential availability of multiple copies of the same message allows also to achieved diversity both in the time and spatial domain. Well known mechanisms such as maximum ratio combining (MRC) and equal gain combining (EGC) can be implemented, as for instance described in [25]. So, cooperation offers different solutions to improve the link performance, as long as nodes synchronization and signal processing are managed properly [26,27].

## 3. Link Layer Protocols

The link layer of the network stack implements many protocols to accomplish different tasks, with the most important being related to error control and channel access. Table 2 summarizes the most important studies performed in this field, focused on the paradigm of cooperative UWSNs. Since providing a reliable data transfer is essential, an efficient check of received information integrity is required. In the case where there is no feedback link between destination and source nodes, the link layer packets gather additional overhead to let the receiver perform forward error correction (FEC) whenever the information is detected as corrupted.

On the other hand, when a bi-direction link is available between nodes, ARQ protocols are considered to guarantee the correct transmission of information. ARQ relies on the use of feedback signaling from the destination to the source to acknowledge whether data delivery was successful or not.

In this latter case, the source retransmits the erroneous packet until its correct reception is achieved. There exist many ARQ implementations, including Stop and Wait (SW-ARQ), Go-Back-*m* (GBm-ARQ) and Selective Repeat (SR-ARQ) [28].

**Table 2 sensors-24-04248-t002:** Summary of link layer protocols and strategies presented in the literature.

Authors	Year	Protocol/Strategy Type
Lee et al. [29]	2010	Cooperative SW-ARQ (relay location aware)
Kim et al. [30]	2016	Cooperative SW-ARQ (relay location aware)
Kim et al. [31]	2018	Cooperative SW-ARQ (relay location aware)
Jamshidi [32]	2019	Cooperative SW-ARQ (relay location unaware)
Gao and Jiang [33]	2012	Cooperative JSW-ARQ
Ghosh et al. [34]	2013	Cooperative HARQ
Goutham and Harigovindan [35]	2021	Adaptive Cooperative HARQ
Goutham and Harigovindan [36]	2023	Cooperative HARQ
Chen et al. [37]	2019	Multi-hop DCC
Khan et al. [38]	2019	AF-based TDMA
Cerqueira et al. [39]	2018	Cooperative SR-ARQ-based TDMA
Rahmati et al. [40]	2019	Cooperative HARQ-based CDMA
Goutham and Harigovindan [41]	2021	DF-based NOMA
Yun [42]	2022	Cooperative-Cognitive channel resource management

The major drawback of ARQ is that packet retransmission is time consuming, especially in the UWSN scenario where the low sound speed and the potential long distance between source and destination may lead to large propagation delay lowering the throughput. In this regard, the cooperation among nodes has been investigated to mitigate such problem. The authors in [29] implement a cooperative SW-ARQ scheme where, together with a source and destination nodes, several intermediate nodes are involved. Given the position of such relay nodes as known to the destination, a cooperative region is defined. So, when the destination node receives a corrupted packet transmitted in a broadcast fashion from the source, it asks first for retransmission from the relay nodes belonging to the cooperative region. A reference description of the cooperative network scenario is given in Figure 4. This strategy reduces the propagation delay related to retransmissions, since relay nodes are expected to be closer to destination than the source node. In [30], a further investigation of the same cooperative ARQ mechanism is provided, by discussing the handshaking procedure to let source and destination nodes identify the cooperative region and the corresponding relay nodes. Finally, a more detailed performance analysis related to throughput, latency and energy efficiency is proposed in [31].

Still related to cooperative SW-ARQ, the authors in [32] define a particular feedback signaling mechanism that allows the relay nodes retransmission scheduling to be managed through a time table. This makes cooperation exploitable without any knowledge about nodes position. Juggling Stop and Wait ARQ (JSW-ARQ) is a modified version of conventional SW-ARQ, that is further adapted to the context of cooperative networks in [33]. In the considered scenario, the source node transmits a sequence of packets to a relay node that, after a correct detection, forwards the packet the destination. By assuming an omnidirectional transmission, the packet forwarded to the destination is received back also by the source and interpreted as a positive reception feedback. In this way, the presence of the relay nodes facilitates not only the packet delivery to the destination, but reduces the typical propagation delay rising due to feedback signaling from the destination to the source. As a result, the transmission efficiency is improved.

Sometimes, when the link layer packet size is too large, retransmitting an entire block of information may be costly and inefficient. So, Hybrid ARQ (HARQ) strategies are considered to merge the advantages brought by FEC and ARQ [43]. Several works have also investigated the reshaping of HARQ in cooperative UWSN scenarios to achieve performance enhancements. In [34], the authors propose a retransmission protocol where the principles of cooperative ARQ are merged with incremental redundancy HARQ. By resorting to rate-compatible punctured convolution codes, it is possible to achieve not only throughput improvement, but also a higher energy efficiency for both source and relay nodes. Cooperative HARQ is also considered in [35], where the destination closest nodes are involved in packet retransmission procedure. Furthermore, the authors propose an optimization algorithm aimed to adapt the modulation scheme and packet size based on the transmission distance between end nodes, thus maximizing nodes energy efficiency with respect to channel conditions. Further analysis of cooperative HARQ is performed by the same authors in [36], where Reed-Solomon coding and selective packet retransmission are considered to improve energy efficiency.

Cooperative ARQ schemes are typically investigated in a single-hop scenario where the relay node is assumed to directly receive the data packet from the source and forward it to the destination. A broader study on the use and impact of retransmission mechanism is instead reported in [37], where Dynamic Coding Cooperation (DCC) is implemented to reduce the end-to-end delay effectively and achieve higher energy efficiency performance with respect to conventional cooperative ARQ and HARQ protocols.

The effectiveness of physical layer techniques and error control based on ARQ can be evaluated also in terms of minimization of channel occupancy due to retransmissions, implicitly resulting as beneficial for channel resources management in a multi-node communication context. In general, channel access in handled by resorting to either orthogonal strategies exploiting time, frequency, code and spatial domains, or non-orthogonal schemes [44]. The authors in [38] consider Time Division Multiple Access (TDMA) performance improved by resorting to relaying nodes that, through AF performed at physical layer, realize spatial diversity to achieve a more reliable signal detection by the destination node. The improvement of TDMA through nodes cooperation is also addressed in [39], where a novel a COoperative Protocol for PERvasive Underwater Acoustic Networks (COPPER) is proposed. Specifically, during each channel access time slot, idle nodes support the active ones by relaying a copy of the transmitted signal towards the destination. The result is twofold, since spatial diversity at physical layer is realized to make signal detection more reliable at the receiver. Furthermore, in case of erroneous link layer packet detection, a cooperative SR-ARQ protocol is activated, with retransmissions being performed by the closest relay nodes and not by the source. By conveniently ruling the packet transmission timing, collision-free access is guaranteed with improved performance in terms of packet error rate, throughput and energy efficiency. In [40], cooperative HARQ is applied to support the reliability performance improvement in Code Division Multiple Access (CDMA). In the proposed mechanism, if the channel between source and destination has poor quality, the transmitting node performs piggybacking on the neighboring nodes with better channels available. This allows the number of packet retransmissions to be reduced, finally improving the system throughput while providing a robust communication. The use of cooperative approaches is also considered in non-orthogonal multiple access (NOMA), as for instance in [41]. Here, the presence of a relaying node performing DF is exploited to realize spatial diversity. So, the destination benefits from multiple copies of the received signal to perform the detection of simultaneously transmitted symbols by resorting to Successive Interference Cancellation (SIC). The reliability performance are investigated in the cases of perfect and imperfect channel knowledge, which plays a fundamental role in SIC operations. Finally, a mechanism for channel resources assignment is proposed in [42], where an underwater network composed of both cognitive and non-cognitive nodes is considered. The cooperation regards the capability of cognitive nodes to perform channel sensing and sharing the related information with non-cognitive nodes, in order to achieve an overall optimized channel usage.

Overall, nodes cooperation for channel occupancy management and error control purposes allows to mitigate another main problem related to underwater acoustic propagation, that is the long propagation delay. However, nodes synchronization is a crucial aspect to deal with at the link layer as well. In fact, in a non-cooperative case, error control performed with ARQ or HARQ involves only source and destination, hence the timing of packet transmission and feedback is managed between a single pair of nodes. On the other hand, in a cooperative scenario with the presence multiple relay nodes, the forward and feedback signaling becomes more complex and, as a consequence, perfectly coordinated transmissions are required. Moreover, an efficient channel resources assignment among the different communication hops may help to further improve the efficiency of nodes cooperation.

## 4. Network Layer Protocols

Typically, underwater nodes are battery-supplied devices that work with low transmit power to reduce the energy consumption. This unavoidably limits the achievable communication range since the acoustic signal may be strongly corrupted by noise and other channel impairments. Therefore, cooperative networking is considered as a promising strategy to deal with such issues since, by means of nodes cooperation, long and unstable links can be replaced with shorter multi-hop routes. A potential reference structure for multi-hop a UWSN scenario is that one provided in Figure 1. However, it is worth highlighting that, while in computer-like networks routing is essentially aimed to optimize data traffic, in UWSNs path selection is more oriented to energy efficiency. In fact, selecting the highest quality network hop reduces the need for potential packet retransmission (at link layer level), hence minimizing the nodes power consumption. So, as already discussed for link layer protocols, even routing strategies at the network layer address challenges rising directly from signal propagation.

One of the main concerns addressed by the literature refers to network clustering, aimed to achieve a convenient nodes grouping and optimize signal path selection. In [45], the authors present a tree-topological based UWSN where nodes exploit packet flooding to get continuously updated information about neighbors availability, so that packet routing and reconfiguration can be performed efficiently and adaptively with respect to the current network status. In [46], nodes clustering is instead realized by resorting to the *k*-means algorithm. Furthermore, by resorting to AF and MRC at the physical layer, a reliable data transfer is provided that, in general, leads to network energy efficiency and throughput improvement. A clustering algorithm oriented to energy optimization is also proposed in [47] for multi-hop network scenarios, where nodes management and path selection are performed based on several information, including the number of neighbor nodes, the nodes residual energy and their distance from the reference sink node. Following a similar approach, a cost function taking into account nodes distance and residual energy is considered in [48] to realize an optimal clustering and routing. Spatial diversity is also exploited at the physical layer to increase the signal detection reliability.

Several works in the literature investigate routing by referring to network scenarios with nodes location awareness, hence path selection is performed essentially based on channel conditions. An example is given in [49], where a multi-sector nodes deployment and sink mobility are considered to handle an energy efficient data routing, supported by cooperative DF to improve the network reliability. Channel awareness also exploited in [50,51], where a cross-layer strategy involving routing relays for packet forwarding and cooperative relays for signaling is proposed. The first ones are selected based on the link capacity, while the second ones are suitably chosen to conveniently realize DF and mitigate the signal propagation impairments. Network nodes clustering and sink mobility are jointly considered in [52]. Here, a sink node operates data collection from sensor nodes located in a specific network area, by eventually resorting to physical layer cooperation in order to improve the link robustness to errors, and forwards the information to the destination node. So the presence of a mobile sink avoids the use unreliable direct links between end nodes, thus improving nodes lifespan, throughput and delay performance. Other approaches to underwater routing consider nodes location unawareness, as for example in [53]. The authors propose in fact two energy efficient routing protocols where path selection is performed based on nodes residual energy, number of hops and the bit error rate measured on the link. This latter features, combined with spatial diversity achieved with nodes cooperation at physical layer, allows not only energy saving, but also performance enhancements in terms packet delivery ratio. Specifically related to a vertical underwater network scenario, the authors in [54] investigate a routing strategy driven by the knowledge about depth of the sensor nodes. By characterizing each node with a location value that is function of depth, routing is performed by selecting those nodes with the lowest value, so to guarantee that the information flows towards the water surface where the destination is supposed to be located. A vertical underwater network routing scenario is also considered in [55], where the authors assume the presence of mobile sink nodes exploitable to realize incremental cooperative routing, with the goal to minimize the nodes energy consumption.

Interestingly, many works in the literature regard opportunistic routing. In opportunistic routing, the source identifies first a set of potential relay nodes, ordered following specific priority rules. So, the relay characterized by the highest priority forwards the packet received by the source to the next-hop node. The other nodes instead pause their other transmission for a certain period, and perform packet retransmission whether the highest priority node transmission fails. The authors in [56] propose three opportunistic pressure based routing mechanisms based on a greedy algorithm to achieve energy saving and minimize the number of transmission hops. Furthermore, relay cooperation is considered to realize spatial diversity and apply MRC at the destination node, in order to improve the communication reliability as well. In [57], opportunistic routing is applied in a vertical underwater network, where path selection is performed based on a depth fitness factor characterizing each node. Such factor accounts for different features like node energy, link distance and packet delivery probability. A fuzzy logic-based relay selection is considered in [58] to realize opportunistic routing, with the best relay node being identified by evaluating the nodes energy consumption and packet delivery success.

As outlined at the beginning of this section, routing in UWSNs is mainly oriented to energy efficiency and/or throughput optimization, without any specific focus on traffic management that typically characterizes other terrestrial networks. Differently, the work in [59] introduces a particular approach where both energy efficiency and traffic optimization are jointly addressed. In fact, the authors investigate a novel routing mechanism based on ant colony algorithm and cooperative relaying, where path next-hop is selected based on both nodes residual energy and transmitted data priority. Merging such features allows routing performance to be improved in terms of network lifetime and load.

The studies about cooperative UWSN clustering and routing discussed above are summarized with their key features in Table 3. It is worth recalling that, differently from what happens in terrestrial RF systems, UWSNs management at the network layer is strictly influenced by the challenges posed by underwater signal propagation. An efficient routing and energy saving depend also on what happens below the network layer. Therefore, the major challenge is UWSNs probably regards the formulation of a cross-layer protocol aimed to harmonize the nodes cooperation at the different levels, so as to maximize the overall network performance in terms of energy and traffic management.

## 5. Numerical Results

In this section, we provide some numerical results obtained through simulations performed with Matlab Software version 2024a, aimed to investigate the performance of the most widespread techniques and protocols exploited in cooperative UWSNs and discussed in the previous sections. Specifically, performance analysis regards physical layer cooperative schemes, link layer ARQ protocols and network layer routing strategies. For the sake of clarity, we would highlight that the goal of such analysis is not to introduce and evaluate the effectiveness of novel solutions, but to support our literature review with some numerical references.

### 5.1. Physical Layer Cooperative Communication Performance

The first part of the analysis concerns the bit error rate (BER) performance considering a communication scenario like that one in Figure 3, including source, destination and a cooperative relay node. The link distance between source and destination has been set to 300 m, with the transmission being performed according to binary phase shift keying (BPSK). Regarding the relay position, we consider three cases, described as follows:Case A: relay located 100 m from the source and 200 m from the destination;Case B: relay located at half way, that is 150 m from the source and 150 m from the destination;Case C: relay located 200 m from the source and 100 m from the destination.

In Figure 5, we compare the performance of AF and DF mechanisms for the mentioned collaborative relay scenarios. Specifically, for both AF and DF we simulated the transmission of a $10^{6}$ BPSK symbols over a noisy channel. The carrier frequency has been set to 20 kHz and the transmission bandwidth to 12 kHz, since representing typical working parameters of commercial acoustic modems and considered in other works [60]. Each node transmit power is set to 1 W. Regarding AF, the signal received by the relay node is first band-pass filtered to remove the noise component out of the signal bandwidth. Then, before forwarding, the physical signal is amplified to match with the maximum amplitude dynamics allowed by the node transmit power. Finally, at the destination node, the received signal is once again first band-pass filtered. Demodulation is performed based on well known matched filtering, and the decision on the received symbol is taken following the maximum likelihood criterion. In DF case, simulation is essentially performed following the same steps, even though at the relay node the signal is first demodulated based on matched-filtering and then re-encoded before being forwarded to the destination. At destination, the BER is calculated as the ratio between the number of wrongly decoded bits and the total number of transmitted bits. Note that, with the employed BPSK modulation encoding one per symbol, we have that BER corresponds also to the symbol error rate. We consider different values of nodes transmit power, so BER is measured as a function of the SNR per bit $E_{b}/N_{0}$ referred to the direct link between source and destination.

By observing the curves it is possible to appreciate that, in general, DF outperforms AF. This is due to the fact that the relay node amplifies not only the useful part of the signal, but also the noise component related to the first hop link. So, even though proper filtering may be applied, a residual noise signal is transmitted over the second hop path, lowering the quality of the signal received by the destination node and thus reducing the detection reliability. Regarding DF, it is interesting to note that best performance are achieved in the case B where the relay is placed at half way from source and destination. So, the quality of both hops is good and balanced, leading the detection to be reliably performed by both relay and destination nodes. Figure 5 describes the performance of collaborative communication that considers the detection performed over a unique signal copy received from the relay. This is sufficient to achieve higher reliability than in the case where direct communication between source and destination nodes is performed. In Figure 6, we instead report the BER performance achieved by performing MRC at the destination, exploiting the signal coming from the source and that one coming from the relay node. Typically, MRC-like mechanisms are implemented in the spatial domain considering a RAKE receiver equipped with multiple antennas, each one collecting a different copy of the received signal. In our case, we reasonably assume that, since the direct and secondary paths have different length, the signal received from the source and the relay arrive separated in time. So, we realize a MRC in the time domain where the different copies of the signal are combined after phase synchronization realized via software. In order to realize MRC, we assume channel state information (CSI) as available at the destination, with combining coefficients being calculated as in [61]. As expected, MRC leads to a more reliable detection, allowing AF to reduce the performance gap with DF.

### 5.2. Link Layer Cooperative ARQ Performance

We pass now to detail the performance of link layer cooperative protocols, by specifically focusing on ARQ and HARQ. We still refer to the three-nodes scenario described above, with the relay located at the reference positions A, B and C. The implemented protocols for error control are cooperative SW-ARQ and cooperative SW-HARQ, respectively, that work as follows. The source transmits a single link layer packet to the destination. If detection fails, the destination asks for retransmission first to the a neighbor node, that is the relay, and finally, if necessary, to the source node. In order to rule the timing of feedback and retransmissions, we consider a timeout for each node sending a packet, so that if no feedback is received withing such time interval, the packet is interpreted as wrongly decoded and so automatically retransmitted. The relay node implements DF to perform packet retransmission. The performance are evaluated in terms of link layer throughput, that is function of the potential retransmissions and propagation delays. By referring to a single packet transmission, the throughput is calculated as:$$T=\frac{L(1-\rho )}{T_{SD}+T_{RD}}=\frac{L(1-\rho )}{N_{s}\left(T_{e}+2\tau_{SD}+T_{f}\right)+N_{r}\left(T_{e}+2\tau_{RD}+T_{f}\right)}$$
where *L* is the packet size expressed in bits, ρ is the percentage of packet bits employed for error detection and correction, $T_{e}$ and $T_{f}$ are the packet and feedback emission time, $\tau_{SD}$ and $\tau_{RD}$ are the signal propagation time from source to destination (and vice versa) and from relay to destination (and vice versa). Finally, $N_{s}$ and $N_{r}$ are the number of retransmissions operated by the source and by the relay. Note that $N_{s}\geq 1$ since the source performs at least the very first transmission, while $N_{r}\geq 0$. For simulations, we set L=100 bits and ρ=0.05 in SW-ARQ, while we consider L=110 bits and ρ=0.13 in SW-HARQ since in this latter mechanism the information block gathers also overhead for error correction. Furthermore, we considered $T_{e}=100$ ms $T_{e}=110$ ms in SW-ARQ and SW-HARQ, respectively, while $T_{f}=1$ ms. Finally, $\tau_{SD}$ and $\tau_{RD}$ depend on the distance between the involved nodes. Even for this simulation case, throughput is also measured as a function of the SNR per bit $E_{b}/N_{0}$ referred to the direct link between source and destination. In other words, it is measured for different node transmit power levels.

Figure 7 reporting the performance of cooperative SW-ARQ shows that the presence of a relay node is beneficial for the management of retransmissions since reducing the propagation delay.

In fact, non-cooperative SW-ARQ (so, involving only source and destination) shows poor performance. Furthermore, the highest throughput is achieved when the collaborative relay node is placed quite close to the destination, so that the link towards the destination suffers from lower attenuation and the probability of successful retransmission increases. Similar trends can be observed also in Figure 8 related to collaborative SW-HARQ. It is important to highlight that having low values of $E_{b}/N_{0}$ means a low nodes a transmit power. As a consequence, due to channel attenuation and absorption, the quality of the link may be very poor and the occurrence of retransmission increases. In this case, HARQ outperforms ARQ since, despite the overhead carried within each packet increases the packet emission time, it can help to correct errors during detection and avoid retransmission requests. On the other hand, a high $E_{b}/N_{0}$ means an increasing nodes transmit power, reflecting on a better link quality with the probability of erroneous packet detection lowering. So, in such condition, the performance of SW-ARQ approach that one in SW-HARQ. It is worth noting that the achieved throughput is in the order of hundreds of bits per second, as we consider a transmission rate equal to 1 kbps based on BPSK modulation. By increasing the data rate, for instance through the adoption of a more spectrally efficient modulation, throughput would scale accordingly.

### 5.3. Network Layer Routing Performance

The literature review has highlighted how energy efficiency is crucial in UWSNs. In this direction, we finally present some results related to routing, demonstrating how the presence of collaborative nodes brings benefits in terms of nodes energy saving. Specifically, we refer to a simulation scenario including several intermediate nodes, ranging from 1 to 8, acting as collaborative relays for the transmission of packets from source to destination. Given CSI as available, path selection is performed through exhaustive search of the hop-by-hop link with the highest channel quality. At each hop, a cooperative DF-based SW-ARQ is performed, where retransmission is firstly asked to those neighboring nodes with good channel quality and, if needed, to the transmitting node. All the mechanisms at physical and link layer are implemented as described in the previous subsections. Performance are measured first in terms of energy efficiency, defined as the ratio between the number of packets to be transmitted and the number of packets (including retransmissions) actually generated within the network. We first refer to a normalized transmit power level for nodes, namely $P_{t}=P_{\mathrm{ref}}=1$ W. Moreover, we consider another case with nodes using a transmit power equal to $P_{t}=0.75P_{\mathrm{ref}}=0.75$ W. The results are shown in Figure 9.

An interesting aspect to discuss is related to the fact that cooperative routing with nodes exploiting a lower transmit power provides the highest energy efficiency, especially when the number of available relay nodes grows. The reason of this result is the following. The use of a high transmit power allows in principle the packet to reach the destination with few hops, but with the potential risk of having a higher number of retransmission requests. On the other hand, a lower transmit power for nodes forces routing to be realized based on a larger number of hops, that however follow shorter and more reliable links. Hence, packet retransmission may be reduced, with a corresponding energy saving for nodes. Furthermore, Figure 9 reports the performance of a non-cooperative routing strategy where, at each hop, error control is managed between transmitting and receiving node without any support by relays. So, packet retransmission in ARQ may be more prone to errors due to the larger distance between the involved nodes. As a result, more transmissions are required by the nodes, impacting on energy consumption.

The presented results suggest how an efficient routing passes through the optimization of many different aspects, including power control, multi-hop scheduling and error control. Overall, the use of a cooperative paradigm in the context of UWSNs guarantees improved performance with respect to conventional approaches.

## 6. Challenges and Future Trends in Cooperative UWSNs

The main takeaway from the reviewed literature and the presented results is that utilizing cooperation leads to a substantial improvement in both throughput and reliability compared to the single-path scenario. In UWAC where signal attenuation, scattering, and multipath effects pose significant challenges, cooperative communication and multi-hop architectures can markedly enhance performance. These methods can extend the range of communication, improve data rates, and increase the robustness of the network against node failures or environmental interference.

### 6.1. Designing Cooperative UWSNs for Real-World Applications

As often happens in engineering, the gains achieved with any novel solution are offset by certain losses or increased costs. In the case of cooperative and multi-hop architectures, the system becomes more complex. Such complexity arises from the need for sophisticated algorithms to manage cooperative strategies and the requirement for precise network synchronization to ensure that data is transmitted and received correctly across multiple hops.

To summarize, the cost is identified by the overall increased complexity of the system, including the need for more advanced hardware and software solutions to handle cooperation and synchronization, and the need for more rigorous maintenance protocols. However, these costs can be justified in real-world underwater applications where traditional single-path communication methods fail to provide reliable and efficient connectivity. For instance, in underwater environmental monitoring, sensor networks can significantly benefit from cooperative communication. By using multiple nodes to relay data, the network can cover larger areas and provide more reliable data transmission, even in the presence of obstacles or varying water conditions. Similarly, in underwater robotics systems used for exploration or maintenance of underwater infrastructures, cooperative communication can ensure that data from various robots is aggregated efficiently and transmitted to the control center, enhancing the overall mission’s success. In underwater rescue operations, where timely and reliable communication is crucial, multi-hop architectures can provide the necessary connectivity between divers, remotely operated vehicles (ROVs), and surface vessels. This ensures that critical information is relayed quickly and accurately, potentially saving lives and resources.

In conclusion, while the implementation of cooperative and multi-hop communications in underwater environments introduces additional complexity and costs, the significant improvements in performance and reliability make it a valuable approach for many real-world applications. The enhanced communication capabilities can lead to more effective monitoring, exploration, maintenance, and rescue operations, thereby overcoming the inherent challenges of UWAC.

### 6.2. The Role of Machine Learning in Future Cooperative UWSNs Management

The state of the art about mechanisms and protocols for cooperative UWSNs clearly highlights how the proposed solutions are aimed to address the challenges rising from underwater signal propagation, independently of the protocol stack layer they are related to. Such particular condition where the physical issues have a straight impact on network management characterizes the underwater acoustic scenario, while it is less evident in other terrestrial systems. Here, performance depends not only on how data transmission is handled, but also on the amount of signaling employed for network management. In fact, due to the slow speed of sound and long propagation delay, the use of heavy overhead signaling leads to significant throughput reduction. Furthermore, network optimization is typically driven by CSI that must be reliable and updated in order to achieve good results. Anyway, the sudden variation of the channel conditions and nodes mobility require frequent update of CSI, that means large overhead signaling and higher energy consumption for nodes.

In this regard, machine learning may potentially represent a revolution for UWAC by enhancing the efficiency, reliability, and robustness of data transmission in challenging aquatic environments. In fact, machine learning algorithms address the mentioned UWSNs challenges by learning from the environment and adapting the communication to its dynamic nature. Regarding the physical layer issues, one significant application regards adaptive modulation and coding, where machine learning models predict the optimal settings to maximize the data rate and minimize errors. Additionally, machine learning enhances signal processing techniques, improving the detection and classification of acoustic signals corrupted by background noise. Moreover, machine learning facilitates the development of access techniques for reducing the amount of interference. Furthermore, it can lead to develop intelligent routing protocols in underwater acoustic networks, ensuring efficient data dissemination and reducing latency. By analyzing patterns and predicting network conditions, these protocols can dynamically adjust routes to maintain robust communication links.

Most of the studies in the literature focus on networking aspects such as routing, even though the physical layer components are taken into account as well. For instance, clustering and routing through improved *k*-means and *Q*-learning algorithms are discussed in [62] to achieve a fair energy consumption among nodes and optimize the bandwidth utilization. Propagation delay, minimal residual energy, and collision rate are the metrics considered in [63] to drive reinforcement learning and *Q*-networks algorithm, with the goal to realize an optimal cooperating node selection aimed to improve the network performance in terms of end-to-end delay, reliability and energy efficiency. Avoiding the use of CSI for path selection is considered in many works where contextual bandit approach is applied [64,65]. Here, relay selection is performed by leveraging environmental data in the learning process directly gathered from sensor nodes, including temperature, wind speed and location. By doing so, CSI acquisition and/or estimation is no more required.

Together with convenient path selection, UWSNs performance can be improved by adapting the nodes power based on the estimated environmental conditions. Such strategy is investigated for instance in [66], where the authors exploit a distributed multi-agent reinforcement learning mechanism that, through power optimization, allows the achievement of a balanced trade-off between single link reliability and network performance. Finally, joint power allocation and relay selection are discussed in [67,68], so that energy efficiency and network capacity issues are simultaneously addressed.

In general, further studies are required to investigate the applicability of machine learning-based techniques in the context of cooperative UWSNs. Firstly, it is important to verify the applicability of these mechanisms in the underwater environment, as they may require significant processing capabilities that may not be available to simple battery-powered underwater nodes. Moreover, the processing time may represent another issue. Actually, this latter is negligible in UWAC since the communication is low rate and and affected by long propagation delays. On the other hand, for high speed underwater links as those based on optical wireless communication, the impact of processing time should be carefully evaluated. Furthermore, in the field of reinforcement learning where a sort of digital twin is developed for considering and training all possible situations, characterizing all the possible parameters (wind, seastream, temperature) may be very challenging. In summary, the integration of machine learning into underwater communications holds immense potential to overcome inherent environmental challenges, leading to more reliable and efficient underwater data transmission systems. Such advancement is pivotal for scientific research, environmental monitoring, and maritime security operations. Besides, a lot of care should be devoted at the expense in terms of latency, processing times and computational cost.

## 7. Conclusions

The realization of an efficient UWSN requires the challenges rising from the signal propagation to be properly addressed. In this regard, the paradigm of cooperative communication and networking firstly developed for terrestrial RF systems has been recast in the context of UWAC as well. This paper discusses the main cooperative strategies and protocols adopted in UWSNs to improve the communication reliability, effectively manage channel usage, error control and routing. Hence, our contribution provides a structured perspective that clarifies how cooperation in UWSNs can be fruitfully exploited at different layers of the protocol stack. The literature review, supported by some numerical results, highlighted the broad benefits brought by the use of a cooperative approach, which reflect in significant network performance improvements in terms of reliability, throughput and energy efficiency. The systematic approach here followed to present the state of the art highlights both the results currently achieved by scientific research and those gaps requiring further investigation. In this regard, future works will be focused on machine learning aided UWSNs, representing one of the emerging and attractive topics in the field of UWAC.

## Figures and Tables

**Figure 1 sensors-24-04248-f001:**
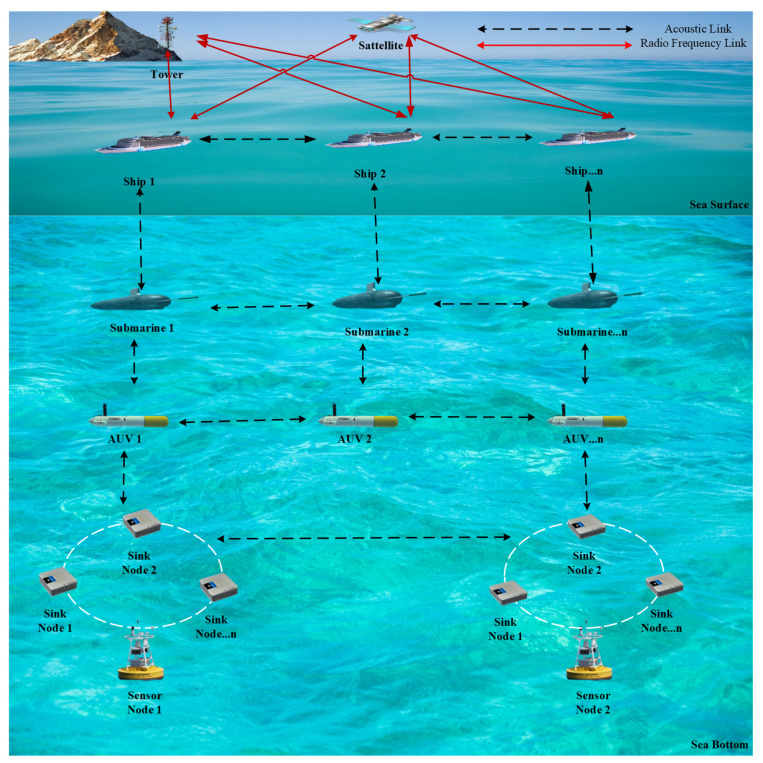
Reference architecture of underwater wireless sensor network.

**Figure 2 sensors-24-04248-f002:**
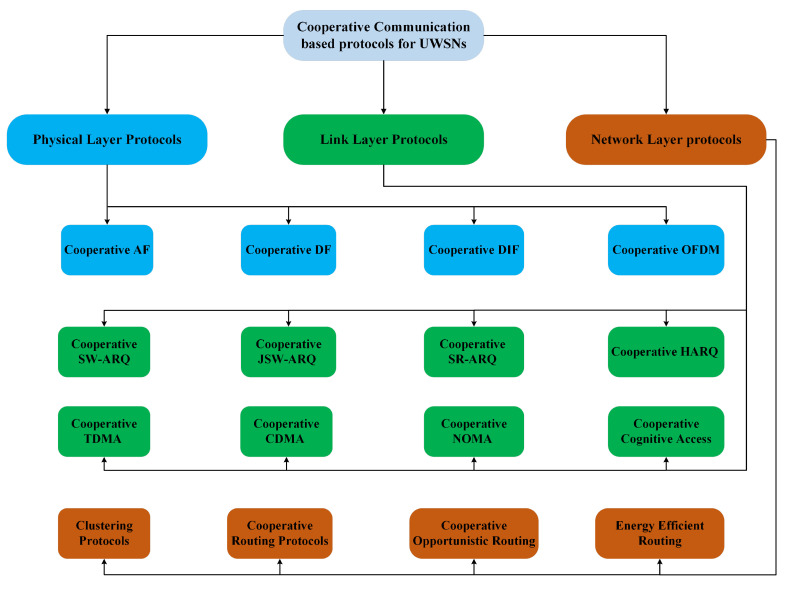
Classification of cooperative communication based protocols for UWSNs.

**Figure 3 sensors-24-04248-f003:**
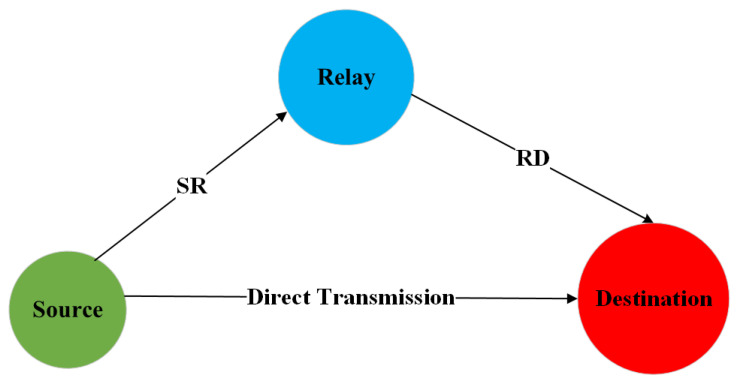
Reference Scenario for Cooperative UWSNs.

**Figure 4 sensors-24-04248-f004:**
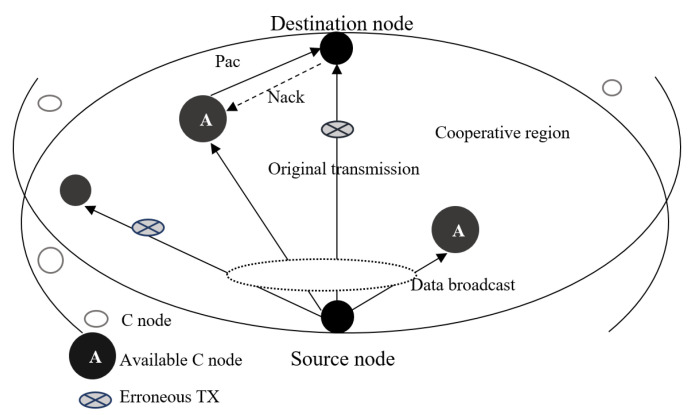
Reference scenario for cooperative ARQ.

**Figure 5 sensors-24-04248-f005:**
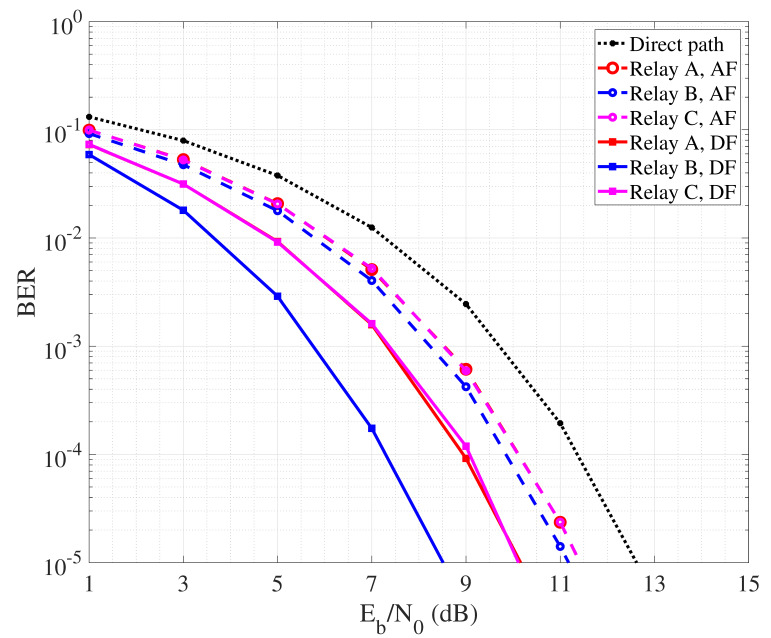
BER performance of cooperative communication with single detection.

**Figure 6 sensors-24-04248-f006:**
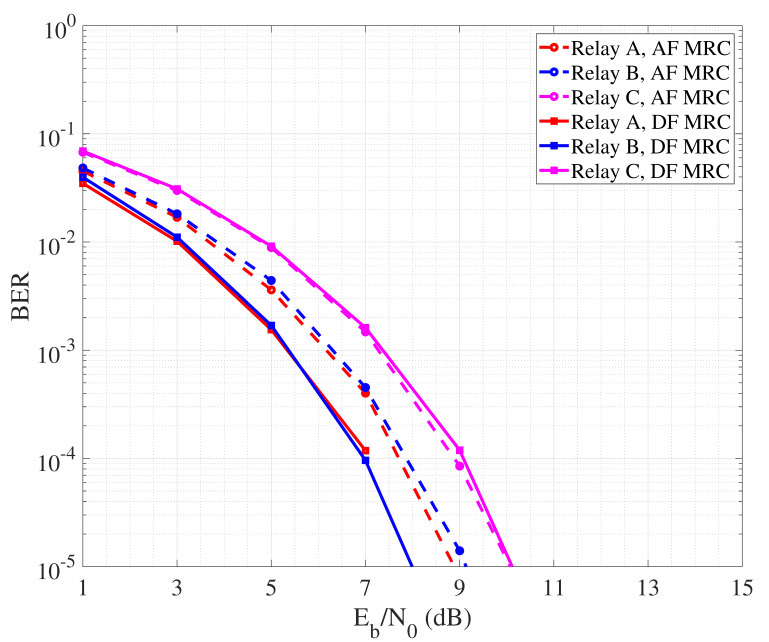
BER performance of cooperative communication with MRC.

**Figure 7 sensors-24-04248-f007:**
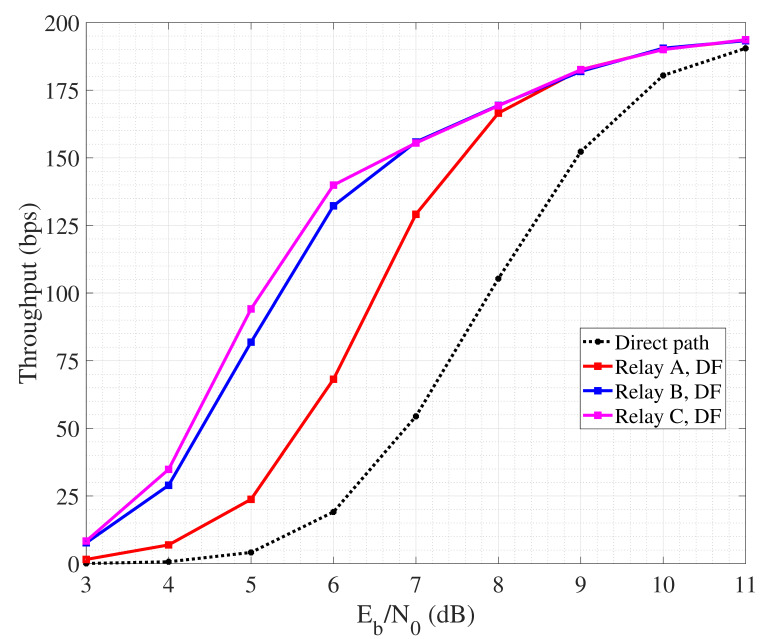
Throughput performance of cooperative SW-ARQ.

**Figure 8 sensors-24-04248-f008:**
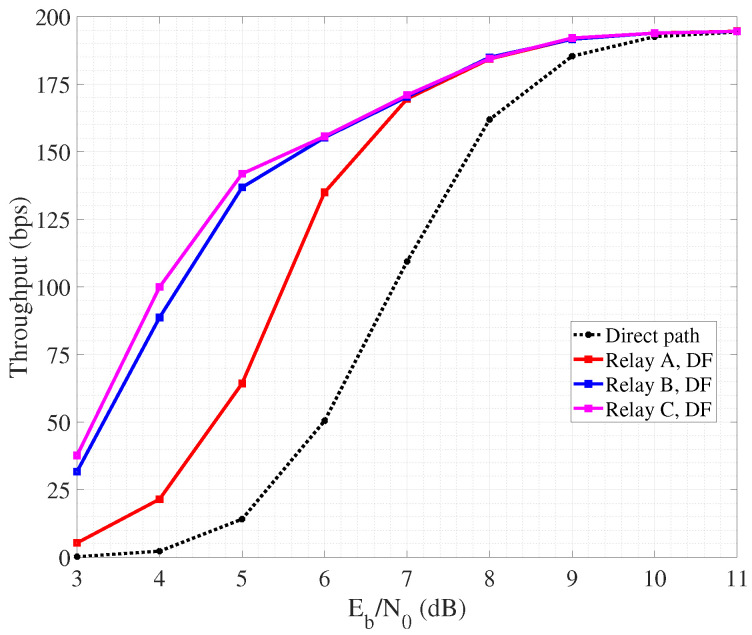
Throughput performance of cooperative SW-HARQ.

**Figure 9 sensors-24-04248-f009:**
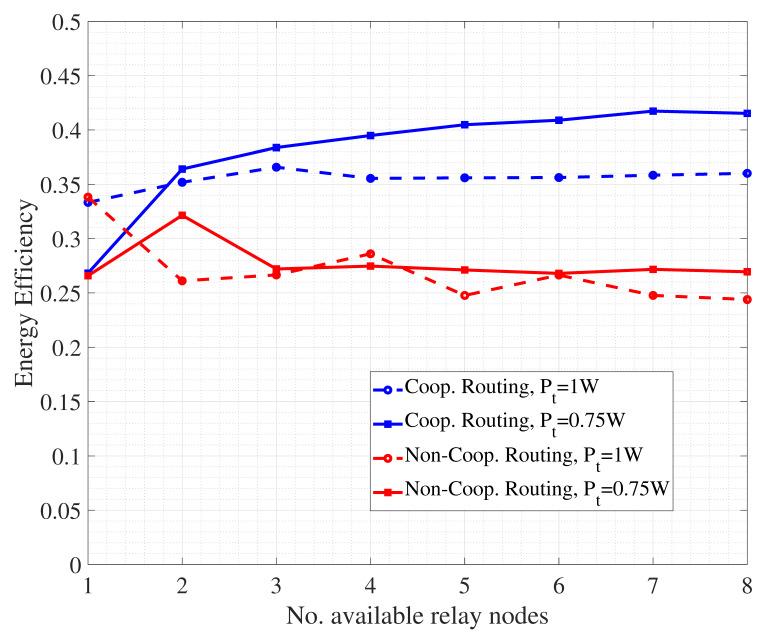
Energy efficiency performance for underwater cooperative and non-cooperative routing.

**Table 1 sensors-24-04248-t001:** Summary of physical layer protocols and strategies presented in the literature.

Authors	Year	Protocol/Strategy Type
Han et al. [16]	2008	Cooperative AF (single-relay)
Vajapeyam et al. [17]	2008	Cooperative AF-STCBC (multi-relay)
Doosti-Aref et al. [18]	2018	Cooperative AF-based OFDM (multi-relay)
Nouri et al. [19]	2016	Cooperative DF (single-carrier)
Huang et al. [20]	2012	Cooperative DF (OFDM)
Liu et al. [21]	2019	Cooperative DIF (single-carrier)
Al-Dharrab [22]	2013	Cooperative AF and DF (OFDM)
Al-Dharrab [23]	2017	Cooperative AF and DF (OFDM)
Wang et al. [24]	2011	Cooperative hybrid AF-DF
Liu et al. [25]	2015	Cooperative AF-based diversity combining

**Table 3 sensors-24-04248-t003:** Summary of network layer protocols and strategies presented in the literature.

Authors	Year	Protocol/Strategy Type
Kim and Cho [45]	2017	Network nodes self-organization
Chen et al. [46]	2020	Clustering and AF-based nodes cooperation
Yu et al. [47]	2020	Energy optimization oriented clustering
Ahmed et al. [48]	2015	Clustering and AF-based nodes cooperation
Ahmad et al. [49]	2022	AF-aided sink mobility based routing
Tran-Dang and Kim [50,51]	2019	DF-TDMA-based channel-aware cooperative routing
Ali et al. [52]	2019	Sink mobility based routing
Shah et al. [53]	2018	AF-aided energy efficient routing (nodes location unaware)
Khan et al. [54]	2018	Depth-based vertical routing (nodes location unaware)
Yahya et al. [55]	2019	Sink mobility based vertical incremental cooperative routing
Javaid et al. [56]	2017	Cooperative opportunistic routing
Karim et al. [57]	2021	Vertical cooperative opportunistic routing
Rahman et al. [58]	2017	Fuzzy logic-based cooperative opportunistic routing
Chen et al. [59]	2021	Data priority and energy efficiency oriented cooperative routing

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
