# Peer review of "Cooperative Communication Based Protocols for Underwater Wireless Sensors Networks: A Review"

_sensors, 2024, doi:10.3390/s24134248_

Round 1
Reviewer 1 Report
Comments and Suggestions for Authors
1. The authors need to rigorously establish the gap in the literature that they attempted to fill in the current review paper.
2. The related work section needs to be expanded, highlighting the pros and cons of the existing works and how the current work has filled any identified gap. A comprehensive Table summarizing the key literature and the gaps being filled should be added accordingly.
3. The authors need to elaborate more clearly on how node cooperation can be exploited at the different levels of the network protocol stack.
4. In Figure 2, the classification of cooperative communication-based protocols for UWSNs has not been adequately described.
5. A section needs to be devoted to discussing the challenges and issues with the deployment of cooperative communication-based protocols for UWSNs.
6. A brief procedure and method for obtaining the results presented should be added to ease results reproduction.
7 A comprehensive section needs to be added on the critical lessons learned/key takeaway lessons from the reviews.
8. Minor English editing is required.
Comments on the Quality of English Language
Minor English editing is required.
Author Response
Please read the attached letter

Reviewer 2 Report
Comments and Suggestions for Authors
The paper addresses a significant topic in the field of UWSN, focusing on cooperative communication protocols. The relevance is high given the growing interest in underwater exploration and monitoring. The paper is well-organized, with a comprehensive literature review, discussion, and a clear structure that guides the reader through the different layers of cooperative communication in UWSNs. It demonstrates a deep understanding of the subject matter.
Here are a couple of minor comments to help improve the paper's flow and clarity:
1. Section 5, Numerical Results: It would be helpful to include an equation or a detailed explanation about how the BER and throughput were determined.
2. Section 5, Numerical Results: The results are presented clearly. Could the authors provide more discussion on what these results imply for real-world applications?
3. Section 6, Future Trends: This section could be enhanced by discussing the potential challenges in implementing machine learning techniques in UWSNs, and how the machine leaching help on these issues, some quantitively comparisons will be great if possible.
Author Response
Please read the attached letter

Round 2
Reviewer 1 Report
Comments and Suggestions for Authors
The future scope of the work should be added at the end of the conclusion.
Comments on the Quality of English Language
Minor English editing is required.
